# G2M Cell Cycle Pathway Score as a Prognostic Biomarker of Metastasis in Estrogen Receptor (ER)-Positive Breast Cancer

**DOI:** 10.3390/ijms21082921

**Published:** 2020-04-22

**Authors:** Masanori Oshi, Hideo Takahashi, Yoshihisa Tokumaru, Li Yan, Omar M. Rashid, Ryusei Matsuyama, Itaru Endo, Kazuaki Takabe

**Affiliations:** 1Department of Surgical Oncology, Roswell Park Comprehensive Cancer Center, Buffalo, NY 14263, USA; masa1101oshi@gmail.com (M.O.); hideo.takahashi@roswellpark.org (H.T.); yoshihisa.tokumaru@roswellpark.org (Y.T.); 2Department of Gastroenterological Surgery, Yokohama City University Graduate School of Medicine, Yokohama 236-0004, Japan; ryusei@yokohama-cu.ac.jp (R.M.); endoit@yokohama-cu.ac.jp (I.E.); 3Department of Surgical Oncology, Graduate School of Medicine, Gifu University, 1-1 Yanagido, Gifu 501-1194, Japan; 4Department of Biostatistics & Bioinformatics, Roswell Park Comprehensive Cancer Center, Buffalo, NY 14263, USA; li.yan@roswellpark.org; 5Department of Surgery, Holy Cross Hospital, Michael and Dianne Bienes Comprehensive Cancer Center, Fort Lauderdale, FL 33308, USA; omarmrashidmdjd@gmail.com; 6Department of Surgery, Massachusetts General Hospital, Boston, MA 02114, USA; 7Department of Gastrointestinal Tract Surgery, Fukushima Medical University School of Medicine, Fukushima 960-1295, Japan; 8Department of Surgery, Jacobs School of Medicine and Biomedical Sciences, State University of New York, Buffalo, NY 14263, USA; 9Department of Surgery, Niigata University Graduate School of Medical and Dental Sciences, Niigata 951-8510, Japan; 10Department of Breast Surgery and Oncology, Tokyo Medical University, Tokyo 160-8402, Japan

**Keywords:** breast cancer, biomarker, cell cycle, gene set, G2M, metastasis, pathway analysis, tumor gene expression

## Abstract

The vast majority of breast cancer death is a result of metastasis. Thus, accurate identification of patients who are likely to have metastasis is expected to improve survival. The G2M checkpoint plays a critical role in cell cycle. We hypothesized that breast cancer tumors with high activity of G2M pathway genes are more aggressive and likely to metastasize. To test this, we used the single-sample gene set variation analysis method to calculate the score for the Hallmark G2M checkpoint pathway using gene expression data of a total of 4626 samples from 12 human breast cancer cohorts. As expected, a high G2M pathway score correlated with enriched tumor expression of other cell proliferation-related gene sets. The score was significantly associated with clinical aggressive features of tumors and patient survival in estrogen receptor (ER)-positive/human epidermal growth factor receptor 2 (HER2)-negative breast cancer. Interestingly, a high G2M score of metastasis tumors was also significantly associated with worse survival. In primary as well as metastasis tumors with high scores, the infiltration of both pro- and anti-cancerous immune cells increased. Tumor G2M score was also associated with treatment response to systemic chemotherapy in ER-positive/HER2-negative cancer, and was predictive of response to cyclin-dependent kinase inhibition therapy.

## 1. Introduction

Currently, there is no cure that can eradicate metastatic breast cancer. In breast cancer, metastasis eventually occurs in 40% of patients, and it is the major cause of the 40,000 breast cancer-related deaths that occur annually in the United States [1]. Thus, precise identification of patients who have a specific treatable target, and who develop metastasis and are at high risk of succumbing to the disease, will have a practical impact on improvement of survival.

Cell cycle progression plays a crucial role in cell proliferation, the alteration of which has been acknowledged as one of the hallmarks of cancer [2]. The G2-M (G2M) checkpoint in cell cycle control mediates G2-M cell cycle transition through cyclin B-cdc2 (CDK1) complexes [3,4], and G2M regulation mediated by CDK1 is a critical factor in tumorigenic events [5]. In recent years, the CDK inhibitors palbociclib, ribociclib, and abemaciclib have emerged as a standard of care in combination with endocrine therapies for patients with estrogen receptor (ER)-positive/human epidermal growth factor receptor 2 (HER2)-negative metastatic breast cancer [6]. ER-positive breast cancer is the most common breast cancer subtype, accounting for approximately 70% of all metastatic breast cancers. Therefore, quantification of the activity of genes involved in the G2M pathway is expected to identify patients who are likely to develop metastasis.

Microarray and sequencing technologies have evolved rapidly in the past few years and revolutionized the collection of DNA and RNA data, making them essential for biomedical research. For example, the computational models involving features like gene expression [7] have been established to predict drug sensitivity for cancer treatment. Yet, it has been reported that findings based on gene-level expression can have limited reproducibility across independent studies. It is also challenging to interpret the biological meaning of changes in the expression of a single gene [8]. Biological phenomena such as drug responses are modulated by the concerted behavior of multiple genes [9], and gene expression scoring capturing multiple genes involved in the same pathway can provide a more accurate value than single-gene scoring [10]. A pathway or gene-sets based approach can take into account such coordination of genes, reduce model complexity, and increase the explanatory power of prediction models [11,12,13]. The single-sample gene set variation analysis (GSVA) computational method can be used to score gene expression at the level of a pathway of interest instead of a single gene, and has been utilized to measure pathway activities in samples [10,14].

In the current study, we hypothesized that the upregulation of the G2M pathway is associated with aggressive cancer biology, a higher likelihood of metastasis, and a worse survival in patients with ER-positive/HER2-negative breast cancer. We used the GSVA method to measure G2M pathway activity of breast cancer tumors, and examined the association of the G2M score with various pathological and clinical features of breast cancer.

## 2. Results

### 2.1. Expression of Cell Proliferation-Related Genes is increased in Breast Cancer with a High G2M Pathway Score

We calculated the G2M scores for 1065 primary tumors of The Cancer Genome Atlas (TCGA) breast invasive carcinoma (BRCA) cohort as the GSVA score for the Hallmark G2-M checkpoint gene set [15]. Tumors were dichotomized into high and low score groups using the median score value. As expected, gene set enrichment analysis showed that tumors with a high G2M score also had enrichment of the expression of genes of multiple gene sets related to cell cycle and cell proliferation. Specifically, in the examination of the 50-member Hallmark gene set collection, significant enrichment was seen for the E2F target, MYC targets v1 and v2, mitotic spindle, MTORC1 signaling, PI3K AKT MTOR, and DNA repair gene sets (false discovery rate (FDR) < 0.05; Figure 1). Because the same gene can be a member of multiple sets, we examined the overlap of genes of these gene sets. The percentage of the genes overlapping with the G2M pathway set was highest for the E2F target (36.5%), but less than 20% for the other gene sets (Appendix A). The association of the G2M pathway score with enriched expression of cell cycle and cell proliferation gene sets that was observed for the TCGA cohort was completely mirrored in a totally independent cohort, the Molecular Taxonomy of breast Cancer International Consortium (METABRIC; Figure 1). These findings suggest that the G2M pathway score reflects cell proliferation in breast cancer.

### 2.2. Increased G2M Pathway Activity in Breast Cancer Tumors is Associated with Worse Clinico-Pathologic Features

Given that the G2M pathway score reflects cell proliferation, we hypothesized that a high G2M score is indicative of aggressive cancer biology, and thus associated with worse clinical outcomes. Indeed, in the TCGA cohort, the tumor G2M score was higher in triple negative breast cancer (TNBC) and HER2-positive subtypes (*p* < 0.001), which are both known to be more aggressive than ER-positive/HER2-negative subtype (Figure 2A; *p* < 0.001). Similarly, the score was significantly higher in tumors with later American Joint Committee on Cancer (AJCC) cancer stage (*p* < 0.001) and greater Nottingham pathological grade as well as the individual Nottingham scores for mitotic count, nuclear pleomorphism, and tubular formation (*p* < 0.001). Strikingly consistent results were seen with the METABRIC cohort (Figure 2A).

TCGA patients whose tumors had high G2M scores had significantly shorter disease-free survival (DFS) as well as disease-specific survival (DSS) compared with patients with G2M pathway low score groups in TCGA cohort (Figure 2C; logrank test, *p* = 0.039 and 0.018, respectively). The association of score with survival was more significant in the METABRIC cohort, which had double the cohort size (Figure 2C; both *p* < 0.001). As the G2M score was higher in aggressive cancer subtypes, as demonstrated in Figure 2A, this survival difference between the groups could reflect the inclusion of aggressive subtypes, such as TNBC, in the high G2M score group. Therefore, we performed survival analyses by cancer subtype. Interestingly, a high G2M score was associated with a shorter survival only in the ER-positive/HER2-negative (Figure 2C; DFS: *p* = 0.043, DSS: *p* = 0.197), but not in the TNBC subtype (DFS: *p* = 0.393, DSS: *p* = 0.226). This result was validated in the METABRIC sub-groups of cases with a stronger significance. For both DFS and DSS among ER-positive/HER2-negative cases, the logrank test *p* was < 0.001, whereas for TNBC, the *p*-values were 0.599 and 0.705, respectively. It is known that low ER expressing tumors demonstrate worse clinical outcomes compared with high ER expressing tumors [16]. To this end, it was of interest whether the association of the G2M score with clinical outcome is confounded by ER expression levels. There was no correlation between the G2M score and ER expression (ESR1 and ESR2 gene expressions for ER-alpha and ER-beta, respectively), which suggests that low ER expression is not confounding the G2M score as it relates to clinical outcome (Appendix A). These findings suggest that a G2M pathway high score reflects aggressive tumor biology, particularly in ER-positive/HER2-negative breast cancer.

### 2.3. Distant Metastasis is More Kikely to Occur in Tumors with High G2M Pathway Activity

With shorter DFS associated with a high G2M pathway score, we hypothesized that tumors with a high G2M score develop recurrence or distant metastasis earlier than those with a low score. In order to test this hypothesis, we identified two independent cohorts with previously published primary tumor gene expression and site-specific metastasis-free survival data (Gene Expression Omnibus studies GSE12276 [17] and GSE2034 [18]). We focused on the lung, brain, and bone as specific metastasis sites as distant metastasis is most common at these sites in breast cancer [19]. Primary tumors with a high G2M score had a significantly shorter overall as well as lung-specific (both logrank test *p* < 0.001), but not for brain- or bone-specific metastasis-free survival in the GSE12276 cohort (Figure 3A). On the other hand, a high G2M pathway score was associated with significantly shorter metastasis-free survival for metastasis to any site (*p* < 0.001), or to the lung (*p* < 0.001), brain (*p* = 0.004), and bone (*p* = 0.036) in the GSE2034 cohort, which has a larger size than GSE12276 (Figure 3B). When examined by ER positivity, a high G2M pathway score was associated with shorter metastasis-free survival only in ER-positive cancer, although the ER-negative sample size was small in GSE2034 cohort, and the ER status of the primary tumors was unavailable for the other cohort (Figure 3B). Thus, the tumor G2M pathway score appears to be useful in predicting distant metastasis among patients with ER-positive breast cancer.

### 2.4. Metastatic Tumors with a High G2M Pathway Score were Associated with Significantly Worse Survival

As tumor metastases are generally more aggressive than their primary tumors, we hypothesized that the G2M pathway activity score would accordingly be higher in metastases. To confirm this, we compared the scores of primary and matched metastasis tumors of the GSE110590 cohort [20]. There were no significant differences in the G2M pathway score among metastases at different sites (Figure 4A) and, contrary to our hypothesis, there was no significant difference in the G2M pathway score between the primary tumors and their matched metastasis for any site (Figure 4A). In this cohort, the basal cancer subtype tended to have higher scores than other subtypes not only for primary tumors, which is consistent with Figure 2, but also in metastatic tumors.

Next, we hypothesized that metastatic tumors with a high G2M pathway score were also associated with a poor prognosis, like the primary tumors with a high score. In order to test this hypothesis, we used the GSE124647 cohort, which includes the gene expression of the various metastatic breast cancers associated with their progression-free survival (PFS). As with Figure 4A, there were no significant differences in the G2M pathway score among the sites of metastatic tumors. Metastatic lesions with a high G2M score were associated with a significantly worse PFS in the whole cohort of metastases as well as sub-groups of local or liver metastases, but not metastases in lymph node, bone, soft tissue, or other metastasis sites. These findings indicate that the G2M pathway activity in tumor metastases also reflects tumor aggressiveness, as it does for primary tumors.

### 2.5. Immune Cell Infiltration is Higher in Tumors with High G2M Pathway Activity

As the tumor immune microenvironment is deeply involved in cancer progression, it was of interest whether primary and metastatic tumors with a high G2M pathway score are associated with unfavorable immune cell infiltration. Using the xCell algorithm on tumor gene expression data of TCGA and METABRIC primary tumor cohorts and GSE110590 and GSE124647 tumor metastasis cohorts, we found that the primary as well as metastasis tumors with a high G2M activity had significantly higher infiltration by both pro- and anti-cancerous immune cells. For primary tumors, in particular, pro-cancerous regulatory T and Th2 helper T cells, and anti-cancerous CD4+ memory T, CD8+ T, Th1 helper T, and M1 macrophage cells were increased in the high-score tumor group of both TCGA and METABRIC cohorts (Figure 5 and Appendix A). For metastasis tumors, Th1 and Th2 T cells were increased in the high-score group of metastases in both cohorts (Figure 5). This was also observed in examinations of site-specific metastases of the two cohorts (Appendix A).

### 2.6. High G2M Pathway Score was Associated with Significantly Better Response to Chemotherapy, but Not with Improved Survival

It is well-known that highly proliferative cancers are more likely to respond to cytotoxic chemotherapy [21]. As we found that a high G2M pathway score reflects cell proliferation, we speculated that the G2M score would decrease when tumors respond to chemotherapy. To examine this, we evaluated data from the GSE28844 cohort [22] in which tumor gene expression data were obtained before and after chemotherapy with doxorubicin. Indeed, the G2M pathway score of primary tumors was significantly reduced by chemotherapy in patients who achieved a good response (t-test *p* < 0.001), while the score did not change in those who had an intermediate (*p* = 0.355) or poor (*p* = 0.226) response (Figure 6A). Furthermore, we found that the G2M score significantly decreased (*p* < 0.001) following successful endocrine therapy with anastrozole and fulvestrant in combination with gefitinib in the GSE33658 cohort [23] (Figure 6A).

As reduction in the G2M score was associated with response to chemotherapy, we further hypothesized that a high score would be associated with pathological complete response (pCR) to chemotherapy. In the GSE25066 cohort [24], we found that the high G2M pathway score prior to the treatment was associated with a significantly higher rate in achieving pCR compared with low score in ER-positive patients (Figure 6B; *p* < 0.001). Although there were no statistically significant differences, there were trends towards a higher pCR rate among the high G2M pathway score groups in ER-positive patients in two other cohorts with very small sample sizes (Figure 6B). Although similar trends were observed in the high G2M pathway score groups among the TNBC patients, it was less prominent and none of them was statistically significant.

As pCR after neoadjuvant chemotherapy is often considered a surrogate marker for better survival, we further hypothesized that a high G2M pathway score was associated with a higher pCR rate and it would also translate into better survival in patients using neoadjuvant chemotherapy. To evaluate this question, we examined the GSE25066 cohort [24]. Disappointingly, the G2M pathway score demonstrated no association with DFS in neither ER-positive or TNBC subtype of cancer (Figure 6C). These findings implicate that while the G2M pathway score may be a potential predictive biomarker for response to neoadjuvant chemotherapy or endocrine therapy, it does not prognosticate improvement in survival owing to therapy.

Given that the G2M checkpoint is an important mechanism of cell cycle control, we assessed if the G2M score is associated with response to CDK inhibitors that directly affect the cell cycle. Because we were unable to identify a CDK inhibitor-treated breast cancer cohort with publicly available gene expression and treatment response data, we examined gene expression and drug response data in the Cancer Cell Line Encyclopedia [25] for five ER-positive/HER2-negative human breast cancer cell lines. The G2M score demonstrated extremely strong correlations with fold-change (FC) and area under the curve (AUC) values for CDK inhibitors (Figure 6D; Spearman r = 1 and 0.783 with *p* < 0.01 and 0.12, respectively). This result suggests that high G2M pathway activity in breast cancer may be predictive of good response to therapy with CDK inhibitors.

## 3. Discussion

We studied a total of 4626 breast cancer tumors for the association between their cell cycle G2M checkpoint activity and with cancer aggressiveness, metastasis, and treatment response. To the best of our knowledge, this is the first study to utilize single-sample gene set expression scoring for elucidating the clinical relevance of the G2M checkpoint.

The G2M activity score was defined as the GSVA score of the “HALLMARK_G2M_checkpoint” gene set [15] using its median as the cut-off. The G2M score significantly enriched expression for cell proliferation-related gene sets such as E2F targets, MYC targets v1 and v2, and the mitotic spindle in GSEA analyses. High G2M score tumor had more aggressive clinical characteristics, such as hormone receptor negative status, higher AJCC cancer staging, and higher Nottingham pathological grade. For ER-positive/HER2-negative breast cancer, high G2M score patients were significantly associated with worse survival. A high G2M score was associated with shorter metastasis-free survival, especially to the lung in ER-positive breast cancer, but not TNBC. A high score was also associated with greater infiltration by both pro- and anti-cancerous immune cells in primary as well as metastatic tumors. The G2M score decreased with a good response to chemotherapy or hormonal therapy. A high G2M score tumor was predictive of a good response to neoadjuvant chemotherapy, but this was not associated with survival benefit. In breast cancer cell-lines, we found that the G2M pathway score had strong correlations with CDK inhibitor cytotoxicity.

As CDK is considered a key molecule for several cell cycle transitions, targeting this pathway has been extensively studied in multiple cancer types during the last decade [26,27,28,29]. CDK1, the only CDK that can initiate the onset of M phase mitosis [30], is essential for the growth of cancer cells as well as normal cells. G2-M regulation mediated by CDK1 has widely been studied for cancer therapeutics in breast cancer [5,31]. Because there are so many molecules involved in the cell cycle, analysis of a single gene expression will not depict the entire picture. In order to overcome this challenge, we used the GSVA to derive a G2M pathway activity score that reflected the activity of 200 genes that are involved in the G2-M checkpoint. As demonstrated in Figure 1, the score represented underlying cell proliferation ability and was associated with other cell cycle gene sets.

The ER-positive subtype accounts for approximately 70% of advanced breast cancer, and is thus responsible for the majority of the deaths from the disease. Identification of the patients who have a higher risks for distant metastasis, especially in the ER-positive subtype, is essential to improve patient survival. While many efforts have been made to predict the response to systemic treatment or the development of metastasis, such as Oncotype DX and MammaPrint [32,33], their utility is still limited. For instance, there is no doubt that Oncotype DX is clinically useful to stratify ER-positive patients to whether they benefit from adjuvant chemotherapy, where now we know that 70% of them do not [33]. However, Oncotype DX only analyzes the expressions of 21 genes, and among them, only 5 genes (Ki67, STK15, Survivin, Cyclin B1, and MYBL2) are proliferation related. This is clearly less accurate in grasping a specific pathway, whereas the G2M score analyzes 200 genes specifically related to the pathway, and thus demonstrates an extremely strong correlation with CDK inhibition. The ability of the G2M pathway score to predict aggressiveness and metastasis shown in multiple cohorts is expected to be useful in patient selection for therapy. This is particularly important in ER-positive breast cancer, where it is well-known that they do not respond to chemotherapy like ER-negative tumors. Thus, a predictive biomarker like the G2M pathway score that identifies aggressive tumors that metastasize followed by a poor prognosis, and that correlate with response to neoadjuvant chemotherapy, is expected to improve efficacy, reduce the toxicity of chemotherapy, and improve patient quality of life. In the present study, the G2M pathway score correlated with the clinical response to both systemic chemotherapy and endocrine therapy.

Unlike in the ER-positive subtype, a high G2M pathway score did not correlate with better response to chemotherapy, nor with worse survival in either primary or metastasis tumors in the TNBC subtype. Given the data that the G2M score was significantly high in TNBC, and the fact that TNBC is biologically aggressive and initially responds to chemotherapy better than ER-positive subtype, we hypothesize that the G2M checkpoint alone is not strong enough to predict the clinical outcome of TNBC tumors.

The utility of the G2M pathway score as a biomarker to predict pCR after neoadjuvant chemotherapy in ER-positive/HER2-negative breast cancer is significantly hindered by the fact that it did not correlate with DFS of patients who underwent neoadjuvant chemotherapy in this study. This result was disappointing because pCR after neoadjuvant chemotherapy is considered a surrogate for a better prognosis. One may argue that this result is because of the small number of ER-positive breast cancer patients who achieved pCR to chemotherapy. However, on the basis of our results that the G2M score correlated with aggressive cancer biology and worse metastasis and survival, it is more likely that cytotoxic chemotherapy was effective enough to achieve pCR in the neoadjuvant setting, but G2M high score tumors were biologically too aggressive for chemotherapy to control the disease in the adjuvant or metastatic setting. A hope is that the G2M score demonstrated strong positive correlation to cytotoxic effect of CDK inhibitors in the breast cancer cell lines that we examined. CDK, which is one of the most critical molecules for several cell cycle transitions, has recently been considered a key target to treat ER-positive breast cancer. Indeed, the results of recent clinical trials indicate that combinations of cell cycle inhibitors and other drugs may be one of the most promising therapeutic approaches to breast cancer in the future [6,34,35,36,37,38]. To this end, we cannot help but speculate that the G2M pathway score may have a utility to be used for patient selection and as a predictive biomarker for CDK inhibitors among patients with ER-positive breast cancer.

Although our findings are novel, our study has a few limitations. First, even though it examined two very large and well-characterized cohorts (TCGA and METABRIC), it remains a retrospective study. A prospective study will be required in order to establish the G2M pathway score as a predictive biomarker. Although we demonstrated a significant association between the G2M pathway score and the effect of CDK inhibitors in human cell lines, which may provide a clue to understanding the interaction between tumors and the response to CDK inhibitors, we were unable to show a similar predictive value of the score for the response to CDK inhibition in patients owing to limited data availability. A randomized clinical trial will be necessary to examine the biomarker utility of the G2M pathway scoring in predicting the effectiveness of CDK inhibitor treatment.

In conclusion, we have demonstrated that the G2M pathway score may serve as a useful tool for identifying patients who are likely to metastasize and have a poor survival in ER-positive/HER2-negative breast cancer. Our findings also support a clinical trial to evaluate the G2M score as a predictive biomarker for response to CDK inhibition therapy.

## 4. Materials and Methods

### 4.1. Data of The Cancer Genome Atlas Breast Cancer Cohort

Tumor gene expression and corresponding clinical data for the TCGA-BRCA project were obtained from the Pan-Cancer Clinical Data Resource [39] and through cBio Cancer Genomic Portal [40], as we previously reported [41,42,43,44,45]. Data for the 1065 female patients were analyzed in the study. As TCGA data do not include information on pathological grade, we used Nottingham histological grade data that had been previously manually collected from pathology reports for 573 of the 1065 patients using software from Text Information Extraction System Cancer Research Network [46]. The Nottingham histological grade is the sum of individual scores for three parameters, degree of tubular formation, nuclear pleomorphism, and mitosis [47].

### 4.2. Data of METABRIC and Other Breast Cancer Cohorts

Normalized microarray-based tumor gene expression and clinical data for 1903 patients of METABRIC cohort were obtained through the cBio portal, as we previously reported [48,49,50]. For the following studies, normalized microarray-based tumor gene expression and clinical data were obtained from Gene Expression Omnibus (GEO) repository (http://www.ncbi.nlm.ih.gov/geo/): Shi et al. (accession number GSE20194; *n* = 278) [51], Symmans et al. (GSE25066; *n* = 508) [24], Vera-Ramirez et al. (GSE28844; *n* = 33) [22], Noguchi et al. (GSE32646; *n* = 115) [52], and Massarweh et al. (GSE33658; *n* = 11) [23]. We used metastasis cohorts (Wang et al. GSE2034 [18]; *n* = 286, Bos et al. GSE12276 [17]; *n* = 204, Siegel et al. GSE110590 [20]; *n* = 16, Sinn et al. GSE124647 [53]; *n* = 140), which have the data of metastatic tumors, to investigate the G2M pathway scores in metastatic tumors. The GSE110590 cohort contained 16 samples of the primary tumors as well as 62 samples of the metastatic tumors from patients with matching primary tumors. The GSE124647 cohort had 140 sample data about metastasis tumors. We used five sample data of ER-positive/HER2-negative breast cancer cell lines with parvo pharmacological profiles obtained from the GSE36139 cohort [25]. When necessary, probe-level microarray data for genes with multiple probes were summarized using mean.

### 4.3. Gene Set Expression Analyses

Log_2_-transformed normalized gene expression data were used. The gene set variation analysis (GSVA) method [14] was utilized to obtain a GSVA score from gene expression data for the “HALLMARK_G2M_checkpoint” gene set of the Molecular Signatures Database Hallmark gene set collection [15]. GSVA Bioconductor package (version 3.10) was used. Within each cohort, tumor samples were categorized into high and low G2M activity score groups using the median GSVA score as cut-off. For gene set enrichment analysis (GSEA) [54], GSEA software (Java version 4.0) and the Hallmark gene set collection were used, as we described previously [55,56,57,58,59,60]. As recommended for GSEA, a false discovery rate threshold of 0.25 was used to deem significance.

### 4.4. Other

Immune features of tumors such as composition of infiltrating immune cells were estimated from whole tumor gene expression data using xCell algorithm [61]. All statistical analyses and data plotting were performed with R software (version 3.6.2) and Microsoft Excel (version 16 for Windows, Redmond, WA, USA). Unless noted otherwise, a *p*-value threshold of 0.05 was used to deem significance. One-way ANOVA or Fisher’s exact test were used to determine the significance of differences for groups. The Kaplan–Meier method with logrank test was used for survival analysis. In data visualizations that are presented here, boxplots are of Tukey type, with boxes depicting median and inter-quartile range.

## Figures and Tables

**Figure 1 ijms-21-02921-f001:**
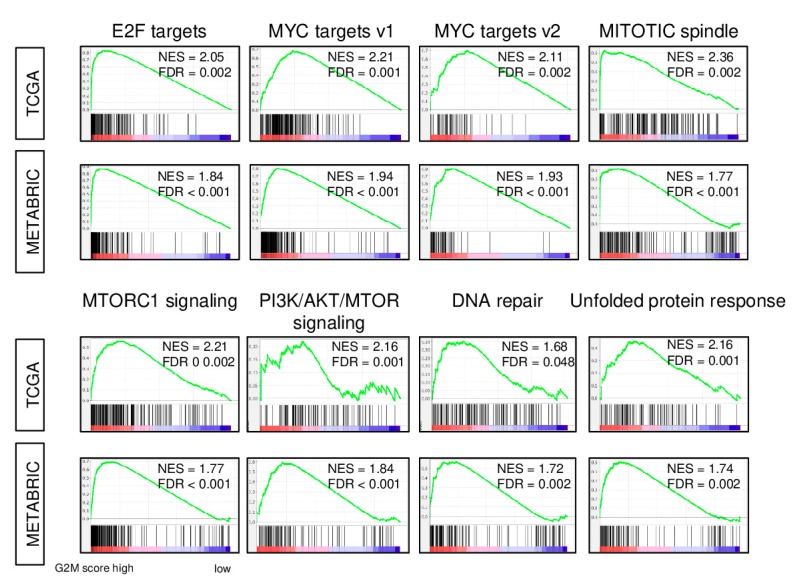
*Hallmark gene sets with a significant enrichment in high G2M pathway score tumors.* Gene set enrichment plots along with normalized enrichment score (*NES*) and false discovery rate (*FDR*) are shown for the eight gene sets for which enrichment was seen in tumors with high compared with low G2M pathway activity score in both The Cancer Genome Atlas (*TCGA*) and Molecular Taxonomy of breast Cancer International Consortium (*METABRIC*) cohorts. The G2M score was calculated from tumor gene expression as the single-sample gene set variation analysis score for the Hallmark G2M gene set, and the within-cohort median value was used to identify tumors with high and low scores. NES and FDR were determined with the classical gene set enrichment analysis (*GSEA*) method.

**Figure 2 ijms-21-02921-f002:**
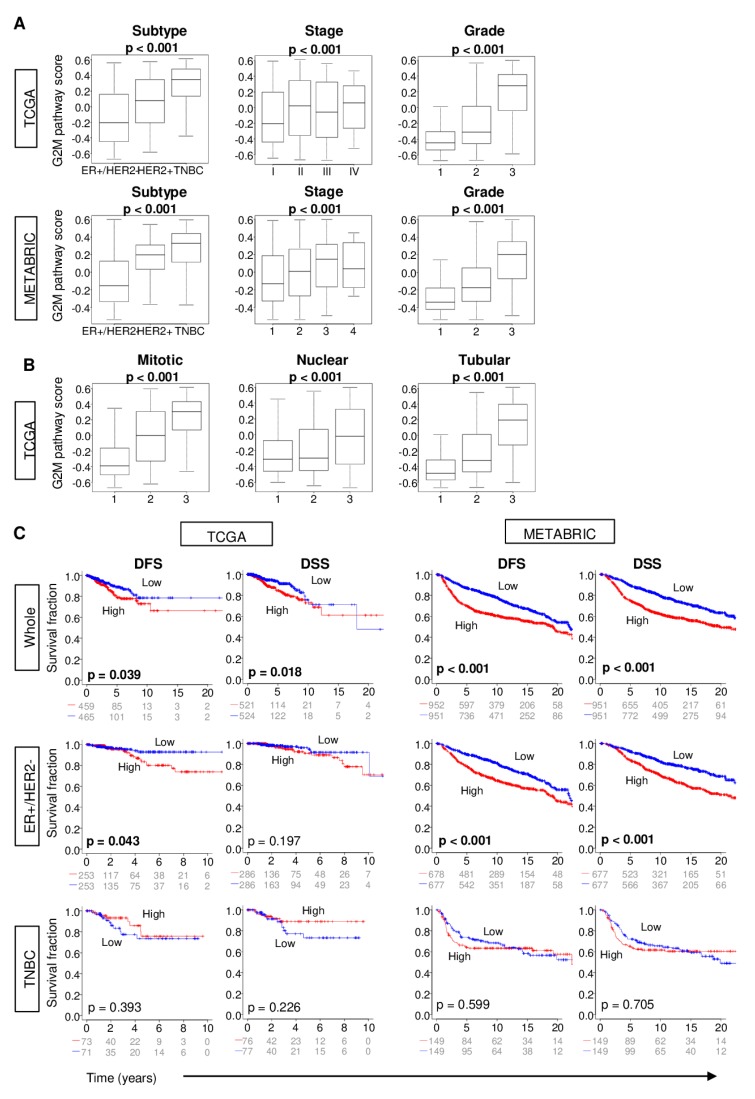
*G2M pathway activity score and tumor characteristics and patient survival.* (**A**). Tukey boxplots depict G2M scores among *TCGA* and *METABRIC* tumors of different grades, American Joint Committee on Cancer clinical stages, and subtypes. (**B**). Tukey boxplots depict G2M scores among *TCGA* tumors of different mitotic, nuclear, and tubular Nottingham histological scores. *p*-values for *A* and *B* are calculated using one-way analysis of variance (*ANOVA*). (**C**). Kaplan–Meier survival plots comparing patients with tumors with high and low G2M scores along with logrank test *p*-values are shown for disease-free (*DFS*) and disease-specific (*DSS*) survival for the entire cohort (*Whole*), or its sub-groups of estrogen receptor (*ER*)-positive/human epidermal growth factor receptor (*HER2*)-negative and triple negative (*TNBC*) patients. The G2M score was calculated from tumor gene expression as the single-sample gene set variation analysis score for the Hallmark G2M gene set, and within-cohort median value was used to identify tumors with high and low scores.

**Figure 3 ijms-21-02921-f003:**
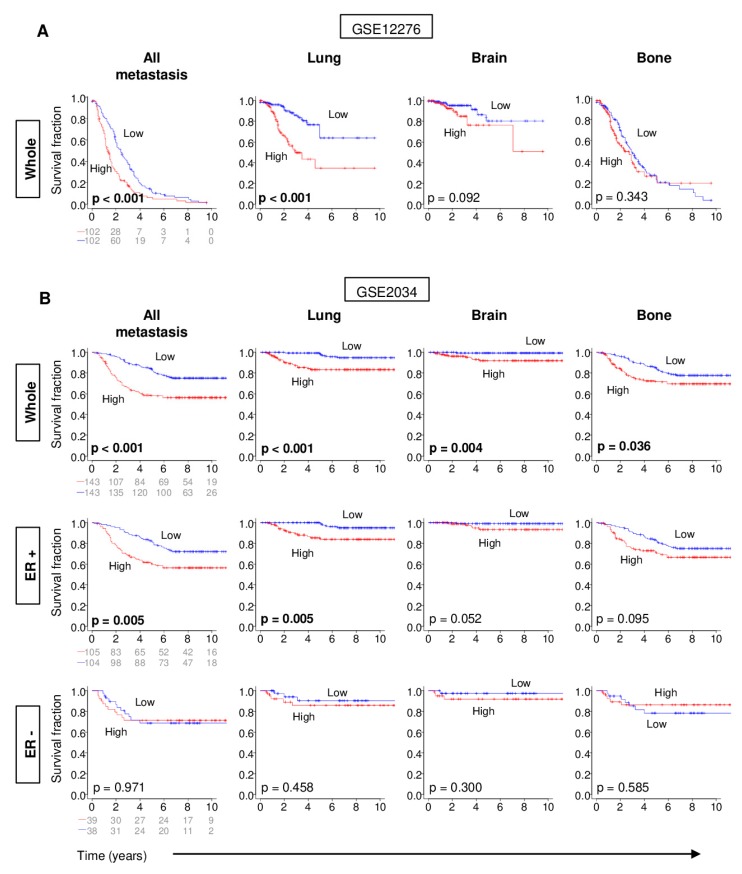
(**A**) The Kaplan–Meier survival plots depicting metastasis-free survival for metastasis to lung, brain, or bone based on G2M pathway score of the primary tumors in the GSE12276 and (**B**) in the GSE2034 cohort. In the GSE2034, metastasis-free survival was demonstrated in the whole cohort, ER+, and TNBC subtypes. The median value of the G2M pathway score was used to divide patients into low and high groups. ER, estrogen receptor; TNBC, triple negative breast cancer.

**Figure 4 ijms-21-02921-f004:**
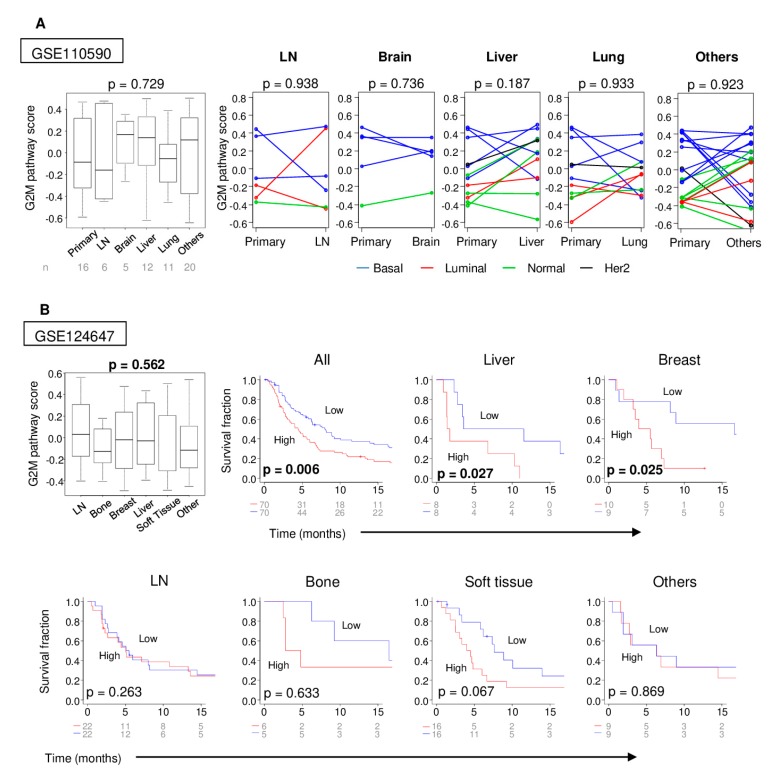
(**A**) Boxplots of the G2M pathway score of primary and each metastatic tumor, and matched pair comparison of G2M pathway scores between the primary and each metastatic tumor, including lymph node, brain, liver, lung and others in GSE110590 cohort. The PAM50 subtypes of cancer are indicated. (**B**) Boxplots of G2M pathway score of primary and each metastatic tumor, and Kaplan–Meier plots with logrank test *p*-values; progression-free survival (*PFS*) between the high- and low-G2M pathway score groups within different metastatic sites in the GSE124647 cohort. The median value of the G2M pathway score was used to divide patients into low and high groups.

**Figure 5 ijms-21-02921-f005:**
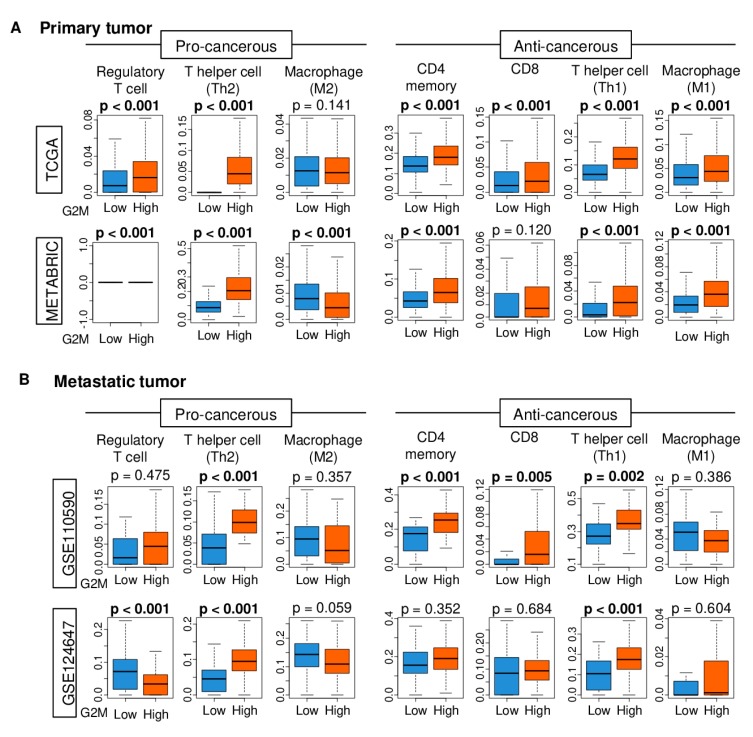
Boxplots of immune cells, CD8 T cell, CD4 memory T cell, T helper cell type 1 and 2, M1 and M2 macrophage, and regulatory T cells depicting the high and low G2M scores (**A**) in primary tumors and (**B**) in metastatic tumors.

**Figure 6 ijms-21-02921-f006:**
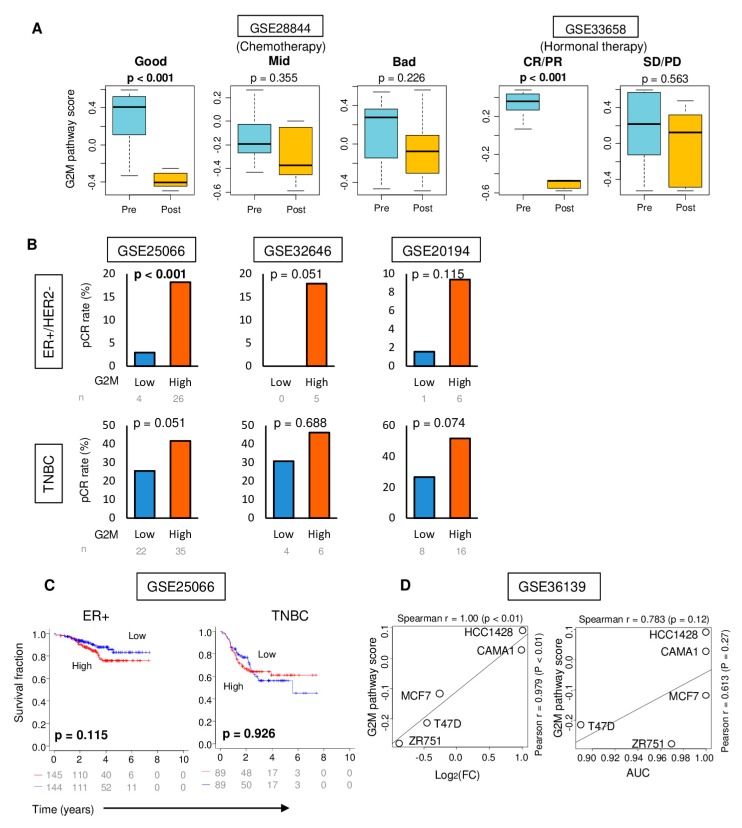
(**A**) Boxplots depicting the scores of pre- and post- treatment using the GSE28844 and GSE33658 cohorts. GSE28844 used neoadjuvant chemotherapy and GSE33658 used endocrine therapy. In GSE28844, response was categorized as Miller–Payne criteria. *RECIST* criteria were used for GSE33658. (**B**) Bar plots depicting the pCR rates between high- and low G2M pathway score groups among patients with ER-positive/HER2-negative tumors and TNBCs in the GSE25066 (*n* = 508), GSE32646 (*n* = 115), and GSE20194 (*n* = 278) cohorts. (**C**) Kaplan–Meier plots with logrank test *p*-values; *DFS* between the high- and low-G2M pathway score groups. ER-positive/HER2-negative and TNBC patients in the GSE25066 cohort. (**D**) Correlation plot between the G2M pathway score and fold-change (*FC*) and area under the curve (*AUC*) using the GSE36139 cohort. AUC, area under the curve; CR, complete response; DFS, disease-free survival; ER, estrogen-receptor; FC, fold-change; pCR, pathological complete response; PD, progressive disease; PR, partial response; RECIST, The Response Evaluation Criteria in Solid Tumors; SD, stable disease; TNBC, triple-negative breast cancer.

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
