# Peer review of "G2M Cell Cycle Pathway Score as a Prognostic Biomarker of Metastasis in Estrogen Receptor (ER)-Positive Breast Cancer"

_ijms, 2020, doi:10.3390/ijms21082921_

Round 1

Reviewer 1 Report

I congratulate the authors for such a descriptive article.

I would like to recommend a little changes in discussion section as in its current form it is very diffused.

Author Response

Reviewer #1 (Remarks to the Author):

I congratulate the authors for such a descriptive article.

Response:

We would like to thank the reviewer for the encouraging remark. 

Comment 1:

I would like to recommend a little changes in discussion section as in its current form it is very diffused.

Response 1:

We agree with the Reviewer 1 that the discussion section of our original version lacked clarity and focus. To this end, we have edited and modified the entire discussion section as below in an attempt to improve its focus and clarity. 

We studied a total of 4626 breast cancer tumors for the association between their cell cycle G2M checkpoint activity and cancer aggressiveness, metastasis, and treatment response. To the best of our knowledge, this is the first study to utilize single-sample gene set expression scoring for elucidating the clinical relevance of the G2M checkpoint.

The G2M activity score was defined as the GSVA score of the “HALLMARK_G2M_checkpoint” gene set [15] using its median as the cut-off. The G2M score significantly enriched expression for cell proliferation-related gene sets such as E2F targets, MYC targets v1 and v2, and the mitotic spindle in GSEA analyses. High G2M score tumors had more aggressive clinical characteristics, such as hormone receptor negative status, higher AJCC cancer staging, and higher Nottingham pathological grade. For ER-positive/HER2-negative breast cancer, high G2M score patients were significantly associated with worse survival. A high G2M score was associated with shorter metastasis-free survival, especially to the lung in ER-positive breast cancer, but not TNBC. A high score was also associated with greater infiltration by both pro- and anti-cancerous immune cells in primary as well as metastatic tumors. The G2M score decreased with good response to chemotherapy or hormonal therapy. A high G2M score tumor was predictive of a good response to neoadjuvant chemotherapy, but this was not associated with any survival benefit. In breast cancer cell-lines, we found that the G2M pathway score had strong correlations with CDK inhibitor cytotoxicity.

Since CDK is considered a key molecule for several cell cycle transitions, targeting this pathway has been extensively studied in multiple cancer types during the last decade [26-29]. CDK1, the only CDK that can initiate the onset of M phase mitosis [30], is essential for the growth of cancer cells as well as normal cells. G2-M regulation mediated by CDK1 has widely been studied for cancer therapeutics in breast cancer [5,31]. Since there are so many molecules involved in the cell cycle, analysis of a single gene expression will not depict the entire picture. In order to overcome this challenge, we used the GSVA to derive a G2M pathway activity score that reflected the activity of 200 genes that are involved in the G2-M checkpoint. As demonstrated in Fig.1, the score represented underlying cell proliferation ability and was associated with other cell cycle gene sets.

The ER-positive subtype accounts for approximately 70% of advanced breast cancer and is thus responsible for majority of the deaths from the disease. Identification of the patients who have a higher risk of distant metastasis, especially in the ER-positive subtype, is essential to improve patient survival. While many efforts have been made to predict response to systemic treatment or the development of metastasis, such as Oncotype DX and MammaPrint [32,33], their utility is still limited. For instance, there is no doubt that Oncotype DX is clinically useful to stratify ER-positive patients to whether they benefit from adjuvant chemotherapy, where now we know that 70% of them do not [33]. However, Oncotype DX only analyzes the expressions of 21 genes, and among them only 5 genes (Ki67, STK15, Survivin, Cyclin B1, and MYBL2) are proliferation related. This is clearly less accurate in grasping a specific pathway, whereas the G2M score analyzes 200 genes specifically related to the pathway, and thus demonstrated an extremely strong correlation with CDK inhibition. The ability of the G2M pathway score to predict aggressiveness and metastasis shown in multiple cohorts is expected to be useful in patient selection for appropriate therapy. This is particularly important in ER-positive breast cancer, where it is well-known that they do not respond to chemotherapy like ER-negative tumors. Thus, a predictive biomarker like the G2M pathway score that identifies aggressive tumors that metastasize followed by a poor prognosis, and that correlate with response to neoadjuvant chemotherapy, is expected to improve efficacy, reduce the toxicity of chemotherapy, and improve patient quality of life. In the present study, the G2M pathway score correlated with the clinical response to both systemic chemotherapy and endocrine therapy.

Unlike in the ER-positive subtype, a high G2M pathway score did not correlate with better response to chemotherapy, nor with worse survival in either primary or metastasis tumors in the TNBC subtype. Given the data that the G2M score was significantly high in TNBC, and the fact that TNBC is biologically aggressive and initially responds to chemotherapy better than ER-positive subtype, we speculate that the G2M checkpoint alone is not strong enough to predict the clinical outcome of TNBC tumors.

The utility of the G2M pathway score as a biomarker to predict pCR after neoadjuvant chemotherapy in ER-positive/HER2-negative breast cancer is significantly hindered by the fact that it did not correlate with DFS of patients who underwent neoadjuvant chemotherapy in this study. This result was disappointing because pCR after neoadjuvant chemotherapy is considered a surrogate for a better prognosis. One may argue that this result is due to the small number of ER-positive breast cancer patients who achieved pCR to chemotherapy. However, based on our results that the G2M score correlated with aggressive cancer biology and worse metastasis and survival, it is more likely that cytotoxic chemotherapy was effective enough to achieve pCR in the neoadjuvant setting, but G2M high score tumors were biologically too aggressive for chemotherapy to control the disease in the adjuvant or metastatic setting. That the G2M score demonstrated a strong positive correlation to the cytotoxic effect of CDK inhibitors in the breast cancer cell lines that we examined offers hope. CDK, which is one of the most critical molecules for several cell cycle transitions, is recently considered a key target to treat ER-positive breast cancer. Indeed, the results of recent clinical trials indicate that combinations of cell cycle inhibitors and other drugs may be one of the most promising therapeutic approaches to breast cancer in the future [6,34-38]. To this end, we cannot help but hypothesize that the G2M pathway score may have a utility to be used for patient selection and as a predictive biomarker for CDK inhibitors among patients with ER-positive breast cancer.

Although our findings are novel, our study has a few limitations. First, even though it examined two very large and well-characterized cohorts (TCGA and METABRIC), it remains a retrospective study. A prospective study will be required in order to establish the G2M pathway score as a predictive biomarker. Although we demonstrated a significant association between the G2M pathway score and the effect of CDK inhibitors in human cell lines, which may provide a clue to understanding the interaction between tumors and the response to CDK inhibitors, we were unable to show a similar predictive value of the score for the response to CDK inhibition in patients due to limited data availability. A randomized clinical trial will be necessary to examine the biomarker utility of the G2M pathway scoring in predicting the effectiveness of CDK inhibitor treatment.

In conclusion, we have demonstrated that the G2M pathway score may serve as a useful tool for identifying patients who are likely to metastasize and have a poor survival in ER-positive/HER2-negative breast cancer. Our findings also support a clinical trial to evaluate the G2M score as a predictive biomarker for response to CDK inhibition therapy.

Reviewer 2 Report

This is a very detailed and scientifically sound study that associates the outcome of ER+ breast cancer (BC) with the expression of the G2M pathway. While the overall findings are very interesting, it nevertheless remains that this is a retrospective study performed on banked material and therefore the results might not necessarily be applicable to the risk startification of patients with BC. If the overall goal of the study is to identify patients with high-risk ER+ positive BC whose tumors express G2M, then there would have to be some commentary involving this approach (i.e., G2M expression) to the more commonly used expression assays (such as Oncotype and Mammaprint). One question: did the authors stratify their analyses according to the level of ER expression, as high ER expressors may have better clinical outcomes compared to low ER expressors, independent of G2M?

Author Response

Reviewer #2 (Remarks to the Author):

This is a very detailed and scientifically sound study that associates the outcome of ER+ breast cancer (BC) with the expression of the G2M pathway.

Response:

We would like to thank the Reviewer 2 for appreciating the quality of our work.

Comment 1:

While the overall findings are very interesting, it nevertheless remains that this is a retrospective study performed on banked material and therefore the results might not necessarily be applicable to the risk startification of patients with BC. If the overall goal of the study is to identify patients with high-risk ER+ positive BC whose tumors express G2M, then there would have to be some commentary involving this approach (i.e., G2M expression) to the more commonly used expression assays (such as Oncotype and Mammaprint).

Response 1:

We totally agree with the Reviewer that our work is a retrospective study and a prospective study will be required in order to confirm the utility of the score. We have added this limitation in the Discussion section as below.

As the Reviewer pointed out, Oncotype and Mammaprint are part of the standard of care. While there is no doubt regarding their clinical usefulness in the stratification of ER-positive patients for adjuvant chemotherapy, the number of genes included in their assay is limited and does not specifically reflect certain pathways. On the other hand, our G2M score analyzed 200 genes specifically related to this cell cycle pathway and demonstrated an extremely strong correlation with CDK inhibition. We have modified the Discussion section as below,

Discussion section:

“Although our findings are novel, our study has a few limitations. First, even though it examined two very large and well-characterized cohorts (TCGA and METABRIC), it remains a retrospective study. A prospective study will be required in order to establish the G2M pathway score as a predictive biomarker.”

“The ER-positive subtype accounts for approximately 70% of advanced breast cancer and is thus responsible for the majority of the deaths from the disease. Identification of the patients who have a higher risk for distant metastasis, especially in the ER-positive subtype, is essential to improve patient survival. While many efforts have been made to predict response to systemic treatment or development of metastasis, such as Oncotype DX and MammaPrint [32,33], their utility is still limited. For instance, there is no doubt that Oncotype DX is clinically useful to stratify ER-positive patients to whether they benefit from adjuvant chemotherapy, where now we know that 70% of them do not [33]. However, Oncotype DX only analyzes the expressions of 21 genes, and among them only 5 genes (Ki67, STK15, Survivin, Cyclin B1, and MYBL2) are proliferation related. This is clearly less accurate in grasping a specific pathway, whereas the G2M score analyzes 200 genes specifically related to the pathway, and thus demonstrates an extremely strong correlation with CDK inhibition.”

Comment 2:

One question: did the authors stratify their analyses according to the level of ER expression, as high ER expressors may have better clinical outcomes compared to low ER expressors, independent of G2M?

Response 2:

We thank the Reviewer 2 for this interesting comment. We agree with the Reviewer that low ER expression may have a worse clinical outcome independent of G2M score, thus, it can be a confounding factor. Although ER status in this study was determined by pathology reports from each cohort, stratification of the patients with ER gene expressions is possible since all the cohorts have comprehensive transcriptome data. We have analyzed the correlation between ER expression (ESR1 and ESR2 gene expressions for ER-alpha and ER-beta, respectively) and G2M score, and no correlation was observed. This means that low ER expression is not confounding the G2M score on clinical outcome. We have added the data in the Result section and Supplementary Figure 1 as the following,

Result 2.2.

“It is known that low ER expressing tumors demonstrate worse clinical outcomes compared to high ER expressing [16]. To this end, it was of interest whether the association of the G2M score with clinical outcome is confounded by ER expression levels. There was no correlation between the G2M score and ER expressions (ESR1 and ESR2 gene expressions for ER-alpha and ER-beta, respectively), which suggests that low ER expression is not confounding the G2M score as it relates to clinical outcome (Fig. S1).”

Supplementary materials:

“Figure S1: Correlation between tumor G2M score and estrogen receptor gene expression.”